# Nondestructive Methodology for Identification of Local Discontinuities in Aluminide Layer-Coated MAR 247 during Its Fatigue Performance

**DOI:** 10.3390/ma14143824

**Published:** 2021-07-08

**Authors:** Dominik Kukla, Mateusz Kopec, Kehuan Wang, Cezary Senderowski, Zbigniew L. Kowalewski

**Affiliations:** 1Institute of Fundamental Technological Research, Polish Academy of Sciences, Pawińskiego 5B, 02-106 Warsaw, Poland; dkukla@ippt.pan.pl (D.K.); zkowalew@ippt.pan.pl (Z.L.K.); 2Department of Mechanical Engineering, Imperial College London, London SW7 2AZ, UK; 3State Key Laboratory of Advanced Welding and Joining, Harbin Institute of Technology, Harbin 150001, China; wangkehuan@hit.edu.cn; 4Department of Materials Technology and Machinery, University of Warmia and Mazury, Oczapowskiego 11 St., 10-719 Olsztyn, Poland

**Keywords:** chemical vapor deposition, nickel alloys, aluminide coatings, fatigue, eddy current

## Abstract

In this paper, the fatigue performance of the aluminide layer-coated and as-received MAR 247 nickel superalloy with three different initial microstructures (fine grain, coarse grain and column-structured grain) was monitored using nondestructive, eddy current methods. The aluminide layers of 20 and 40 µm were obtained through the chemical vapor deposition (CVD) process in the hydrogen protective atmosphere for 8 and 12 h at the temperature of 1040 °C and internal pressure of 150 mbar. A microstructure of MAR 247 nickel superalloy and the coating were characterized using light optical microscopy (LOM), scanning electron microscopy (SEM) and X-ray energy dispersive spectroscopy (EDS). It was found that fatigue performance was mainly driven by the initial microstructure of MAR 247 nickel superalloy and the thickness of the aluminide layer. Furthermore, the elaborated methodology allowed in situ eddy current measurements that enabled us to localize the area with potential crack initiation and its propagation during 60,000 loading cycles.

## 1. Introduction

Nickel superalloys are commonly used in aircraft engines due to their superior, high-temperature performance properties, including corrosion, heat and creep resistance [1,2]. The most conventional nickel superalloys for gas turbines are MAR 247 [3], Rene 80 [4] and IN738 [5]. It should be emphasized that MAR 247 exhibits higher strength properties and better creep response in comparison to conventional alloys for aircraft engine components [3,4,5]. Increased demand for new materials in the aircraft industry led to the fabrication of new generation, cost-effective alloys doped with Re and Ru [6]. However, high costs limit the application of new generation materials in mass production.

The fatigue life of nickel-based superalloys is mainly determined by their initial microstructure, morphology and volume fraction of γ′ and γ″ precipitates [7]. In a comprehensive review presented by Garimella et al. [7], it was reported that ageing accompanied by the precipitation of γ′ precipitates could enhance fatigue properties. It was shown that the crack-growth rates were lower for materials that had been aged to peak hardness compared to the underaged and as-received specimens. Moreover, it was further confirmed that the maximum stress increased with increasing volume of the γ′ participation up to the peak-aged condition and decreased in the overaged condition. It should also be mentioned that materials with the coarse-grained structure exhibit a shorter low-cycle fatigue life than fine-grained ones [7]. Furthermore, finer precipitates resulted in a slower long-crack growth rate and longer low-cycle fatigue life (LCF) of the nickel-based superalloys, such as Inconel 718 [7].

In order to enhance the mechanical properties and service life of commonly used nickel superalloys, the Ni-Al-type intermetallics were used as coating materials. These materials are characterized by a crystalline structure with strong chemical bonds and tightly packed atoms in the lattice, which improve their thermal stability significantly. It was found that only NiAl and Ni_3_Al coatings could form a beneficial coating structure that could transfer high mechanical loads in aggressive environments [8,9]. Ni_3_Al-based intermetallics exhibited improved fatigue strength at high temperatures in comparison to the commonly used Ni-based superalloys [10]. The superior properties of thermally stable NiAl- and Ni_3_Al-based materials at high temperatures could be potentially used as the protective coatings for aircraft engine turbines [11,12]. Generally, thermal barrier coatings are applied to the parts of a gas turbine to reduce their operational temperature by approximately 100–300 °C while simultaneously increasing their service life [13]. The content of aluminum and chromium within the coating allows it to form a stable oxide layer that protects the substrate material and significantly reduces its chemical reactivity during performance in aggressive environments [14,15,16]. Since a high temperature significantly reduces the fatigue life of nickel superalloys [16], the application of thermal barrier coatings could further extend their fatigue life. The high-temperature fatigue response of the coated MAR 247 nickel superalloy has been widely studied in the literature [17,18,19,20]. It should be mentioned, however, that before the engine attains the high working temperature, the turbine blades are subjected to loading at ambient temperature. On the other hand, the nickel-based superalloys used for aero application mainly work at elevated temperatures, and thus, most studies were devoted to testing in aggressive environments. However, before any part will start its high-temperature performance, it should withstand the high load at ambient temperatures up to 300 °C. Nickel-based superalloys exhibit microstructural and mechanical stability at such temperatures, and therefore, the fatigue testing performed at room temperature allows us to characterize their high cyclic load capability. Unfortunately, there are not many available publications devoted to the effect of coating on the fatigue performance of the nickel-based superalloys at ambient temperatures. Increasing demands within the aviation industry for a new generation of turbine blade materials led to the development of new diagnostic techniques. Advanced techniques, such as the blade-surface images analysis [21], optoelectronic and thermographic methods [22], eddy current and ultrasonic methods [23], vibrothermography [24] or even Digital Image Correlation [25], allow for the identification of cracks, intrinsic defects, subsurface defects, pores and potential areas of crack initiation. The variety of destructive and nondestructive methods for nickel-based superalloy inspection allows us to identify potential reasons for the material decohesion. However, a correlation between destructive and nondestructive methods has not been sufficiently reported as yet.

Therefore, the main aim of this work was to assess the effect of aluminide layer thickness and the initial microstructure of MAR 247 on its fatigue performance during in situ eddy current measurements. Such tests enabled the characterization of the mechanical properties of three different initial microstructures and two different thicknesses of NiAl coating obtained by using the CVD process with optimized parameters presented in the authors’ previous paper [26] and assessed the effectiveness of the eddy current method for in situ measurements that allow the identification of areas of potential crack. The specific microstructures and coating thicknesses were used in this paper due to their superior high temperature performance reported by authors in a different paper [27].

## 2. Materials and Methods

Specimens made of MAR 247 nickel superalloy were manufactured using conventional casting process in an ALD vacuum furnace (ALD Vacuum Technologies GmbH, Hanau, Germany). MAR 247 nickel superalloy specimens with equiaxed microstructures (EQ) were quenched with the furnace to achieve the required fine (Figure 1a) and coarse (Figure 1b) microstructure. In order to classify the sizes of equiaxed grains, the cast specimens were thermally insulated using ceramic wool (3 layers for fine-grained and 6 layers for coarse-grained microstructures). Specimens with directional grain orientation (DS) (Figure 1c) were transferred outside the furnace under controlled speeds of 3 mm/min, which enabled the formation of the columnar structure. The chemical composition of MAR247 nickel superalloy was presented in Table 1.

Aluminide coatings were obtained using the CVD process and Ion-Bond setup (Ion Bond Bernex BPX Pro 325 S, IHI Ion bond AG, Olten, Switzerland) located in the Materials Testing Laboratory for the Aviation Industry of the Rzeszów University of Technology, Poland. The optimized CVD parameters were obtained for the hydrogen protective atmosphere, with a deposition time of 8 and 12 h at the temperature of 1040 °C and internal pressure of 150 mbar [27]. The microstructural observations and chemical composition analysis were performed using a Hitachi 2600N scanning electron microscope with an energy dispersive spectroscopy (EDS) attachment (Oxford Instruments, Oxford, UK). The microhardness of the as-received and coated material was determined on a ZWICK hardness tester (Materialprüfung 3212002, Ulm, Germany) using the Vickers method. Standard fatigue tests were performed using the MTS 810 testing machine (MTS System, MN, USA) and the conventional MTS extensometer. Uniaxial tensile tests were carried out at strain rate equal to 2 × 10^−4^ s^−1^ using five specimens. Fatigue tests at a temperature of 23 °C were force controlled under the zero mean level, constant stress amplitude and a frequency of 20 Hz. Every fatigue test was performed at least twice to guarantee the reliability of the results obtained. The range of stress amplitude from 350 to 650 MPa was established on the basis of the conventional yield point R_0.2_ determined from the uniaxial tensile test. The geometry of the specimens is presented in Figure 2. The eddy current measurements were performed using an Olympus Nortec 600 D (Olympus, Tokio, Japan) flaw detector and pencil probes with operating frequencies of 100–500 kHz and 1–6 MHz. The signal was calibrated using a standardized pattern sample with reference electric discharge machined (EDM) notches of 0.1, 0.2, 0.5 and 1 mm depth. Such a sample enables the optimization of the measuring parameters for the best surface and subsurface defect detection. EC measurements were performed on three different areas located in the gauge length and on both sides of the specimen using frequency equal to 240 Hz, after each of 10,000 cycles. Selected values of frequency enabled a penetration for depth of approx. 1 mm for 240 Hz.

## 3. Results and Discussion

### 3.1. Microstructural Characterization of Coating during CVD Process

The microstructure of the MAR 247 nickel superalloy after the CVD process performed at 1040 °C revealed the presence of columnar and equiaxial dendrites, as shown in Figure 3. Based on the quenching conditions applied, different structures were observed. A fine-grained structure (Figure 3a) was achieved during slow cooling with the furnace. It should be mentioned that the static recrystallization temperature of the nickel-base alloys is within the range from 1000 to 1100 °C [28]. During the CVD process, the specimens were exposed to temperatures over 1000 °C, and therefore, a recrystallization could occur. During high-temperature exposure, the deformed and elongated grains would transform into finer equiaxial grains [29]. A coarse-grained structure (Figure 3b) was characterized by large grains, which may indicate that high-temperature exposure led to extensive grain growth. On the other hand, columnar grains (Figure 3c) grew parallel to the solidification direction. They were formed after the casting process, performed under carefully controlled speeds.

The specimen view of 40 µm-thick NiAl coating is presented in Figure 4. Its structure was mainly determined by the growth kinetics and conditioned by the temperature, pressure and different synthesis time in the CVD process. NiAl coating was uniformly distributed on the MAR 247 surface. The cross-sectional view allowed us to distinguish its two-layer structure: a homogeneous zone of secondary solid solution of the β (NiAl) phase and heterogeneous NiAl matrix (dark grey) with Ni_3_Al phase dispersions (bright grey). The lower content of Al and the participation of Co, Cr and Ti alloying elements were found in the interlayer zone. This is due to the atom diffusion from the substrate. The chemical composition analysis was performed in the cross-section of specimens at seven points starting from the edge into the substrate (Figure 4, Table 2). The content of aluminum gradually decreased from 25% on the edge to ~6% within the substrate material. The coating was characterized by the typical intermetallic superstructure of the secondary solid solution β with B2 ordered structure [30]. One can conclude that the CVD process parameters were successfully selected since no defects were observed between the coated material and the coating itself. Minor cracks between sub-layers were caused by extensive grinding during the preparation of the metallographic specimens for the microstructural observations. The detailed microstructural characterization and mechanical properties of the NiAl coating obtained, including hot resistance, adhesion and wear resistance, were presented in the authors’ previous work [26].

### 3.2. Effect of Coating Thickness on Fatigue Behavior

The effect of coating thickness on the mechanical properties of the MAR247 nickel-based superalloy was also investigated in the standard fatigue tests. The results of experiments showed that the coating slightly decreased the stress response of the MAR 247 specimens at temperatures of 23 °C (Figure 5). Based on the analysis of S-N curves, it can be assumed that both fine- and coarse-grained microstructures do not affect the fatigue behavior of the MAR 247 nickel-based superalloy, since similar stress responses were achieved for both of these structures. It should be mentioned, however, that the column grain structure was characterized by the weakest properties in comparison to other structures. It was reported by Sulak et al. [31] that the cyclic deformation of nickel-based superalloys is mainly determined by the interaction between dislocations and γ strengthening phase. This is due to the fact that the precipitates played a dominant role as an effective barrier against the dislocation movement. Regardless of the initial microstructure, it was found that a difference in coating thickness affects the stress response of the MAR 247 nickel-based superalloy. A slight reduction in the stress amplitude could be observed when the thicker coating was applied to the substrate material. Such behavior might be caused by the high hardness and stiffness of the protective coating. Similar findings were presented in [32,33,34], where the fatigue life of steel decreased with increasing coating thickness. Such effect was explained using the linear elastic fracture mechanics under the assumption of small-scale yielding [33]. It was found that the strain energy stored in the coating layer contributed to the crack development. On the other hand, Akebono et al. [32,35] reported that two main factors could affect the fatigue performance of the Ni-based coated steel substrate. The first one was found to be the size and population of porosities in the coating. As the fatigue crack initiation was taking place at the porosities, specimens exhibited lower fatigue strength. The second factor was related to the lowered hardness of the Ni matrix accompanying chromium segregation. The same behavior can be observed in Figure 5, Figure 6 and Figure 7, where the 40 µm-thick NiAl coating affected the fatigue performance more than that of the 20 µm one. It was found that fusion for a shorter time is more effective in producing sprayed materials of better fatigue properties [35].

A comparison of the fatigue response of three different microstructures and two thicknesses of the NiAl-coated MAR 247 nickel-based superalloy is presented in Figure 8 and Figure 9. It was found that the fine-grained MAR 247 nickel-based superalloy can be characterized by the best fatigue response either for the 20 or 40 µm-thick coating. It is widely known that the microstructure refinement slightly affects the fatigue behavior of the nickel-based superalloys [36]. A similar result was reported by Mukhtarov et al. [37], where a 718 nickel-based superalloy with a different grain size (0.1–10 µm) was subjected to the low cyclic fatigue at a temperature of 23 °C. It was observed that the lifetime of the 718 nickel-based superalloy was approximately the same, regardless of the grain size. Therefore, one can conclude that the main factor affecting the fatigue performance of the MAR 247 nickel-based superalloy could be attributed to the coating itself. The results presented in Figure 8 and Figure 9 and those reported by Akebono et al. [35] allowed us to clearly state that coating thickness strongly affected the fatigue properties, and as a consequence, when the thinner coating thickness was applied, a higher fatigue strength was achieved. Such behavior was related to the fatigue crack propagation through many defects on the coated surface in which the number of the coating defects and their sizes were mainly dependent on the coating thickness. For the thicker coatings, a larger number of defects with greater sizes were observed. Consequentially, the coated specimens with thinner coatings indicated a higher fatigue strength.

The eddy currents measurements were performed for specimens tested in the low-cycle fatigue regime (LCF) under four cyclic stress amplitudes ranging from 450 to 650 MPa and a frequency equal to 240 Hz (Figure 10a–c). The measurements were performed in situ in six different stages of the fatigue, i.e., during fatigue tests for every 10,000 cycles and up to 60,000 cycles. When the specimen reached the specific number of cycles, the fatigue test was interrupted, and measurements were performed. After reaching 60,000 cycles, the specimen was tested to failure without subsequent EC measurements. It was found that the electrical conductivity value decreased with the number of cycles to failure, which can be attributed to the gradual degradation of the material during fatigue testing (Figure 11). Such effect was observed for all specimens, regardless of their initial microstructure or the coating thickness. The eddy current method enabled the identification of fatigue cracks at the stage preceding a development of the dominant crack in the period of 60,000 loading cycles. The significant decrease in the electrical conductivity value of about 0.5 indicates a crack initiation in the scanned area. The most important condition for crack identification is a dimension of its depth, which should exceed 50 µm. Therefore, during the in situ fatigue crack diagnostics of structural elements, the EC method was found to be useful in the effective identification of the potential areas of crack initiation (Figure 10c). Nondestructive techniques have been widely used to monitor fatigue cracks. In a comprehensive review prepared by Kong et al. [38], the eddy current method was found to be as promising and more sensitive than many other techniques (X-ray radiography, ultrasound testing, thermography, acoustic emission and digital image correlation) for potential crack localization. The eddy current method is particularly successful in fatigue damage monitoring, since it enables the determination of a number of cycles to the crack initiation [39,40,41]. It was shown by Potthoff et al. [42] that this method could be effectively used for the monitoring of low-cycle fatigue damage of the cast metals. In this study, the next step was taken towards extending this method to damage monitoring in coated materials.

The fracture surface inspections on specimens subjected to high-cycle fatigue tests were carried out. The results are presented in Figure 12. One can indicate that the CVD process parameters were successfully applied as the NiAl coating remained adhered to the substrate after decohesion within the specimen gauge length. Moreover, no particular damage or decohesion between the sublayers was observed in the area close to the dominant propagating crack for all the specimens employed. The occurrence of voids on the fracture surface was not observed either. The fine-grained fracture surface consisted of a developed structure with large and small dimples (Figure 12a,b marked with arrows). Such microstructure reflected a plastic deformation of 8% during uniaxial tensile tests and the best relative fatigue performance. The cracks found within the surface of the coarse structure (Figure 12c,d marked with arrows) were related to the brittleness of the grain boundaries. The crack formation of the nickel-based superalloys at ambient temperatures was mainly determined by the existence of carbide precipitates on the grain boundaries [42]. The occurrence of M_23_C_6_ carbides precipitated at the grain boundaries could further improve the mechanical properties and prevent against grain boundary sliding [43]. Column structured (Figure 12e,f marked with arrows) fracture surfaces were characterized by a striation pattern that is typical for the transgranular fracture. These striation patterns on the fracture surface were probably caused by the presence of large individual grains in the column-grained structure. It was concluded that grain boundary crack formation mainly determines the minimum ductility of the MAR 247 nickel-based superalloy. The cracks might nucleate from both the coating and the grain boundary carbides. However, the area on the coating where the initial crack occurred might be the potential place of crack propagation. The NiAl coating applied did not change the deformation mechanisms of the nickel-based superalloys, which are mainly determined by the deformation conditions and initial microstructure of the alloy investigated.

## 4. Conclusions

The LCF performance of the aluminide layer-coated and as-received MAR 247 nickel superalloy with three different initial microstructures (fine grain, coarse grain and column-structured grain) was monitored using a nondestructive, eddy current method. The fatigue performance was mainly driven by the initial microstructure of the MAR 247 nickel superalloy and the thickness of the aluminide layer. The fine-grained MAR 247 nickel-based superalloy was characterized by the best fatigue response for the 20 or 40 µm-thick coating.

It was also found that the eddy current method could be successfully used to monitor the low-cycle fatigue performance of the coated nickel-based superalloys. The changes in electrical conductivity could be used as an effective indicator of damage development in the structural materials. The methodology proposed enables the in situ eddy current measurements that make it possible to successfully identify the area with potential crack initiation and its propagation within a range of cycles representing LCF.

## Figures and Tables

**Figure 1 materials-14-03824-f001:**
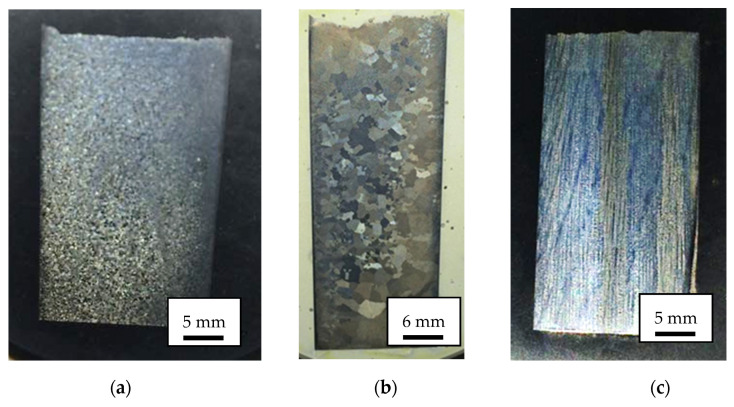
Macroscopic structure of MAR 247 nickel superalloy: (**a**) fine grain, (**b**) coarse grain and (**c**) column grain observed by using light optical microscope.

**Figure 2 materials-14-03824-f002:**
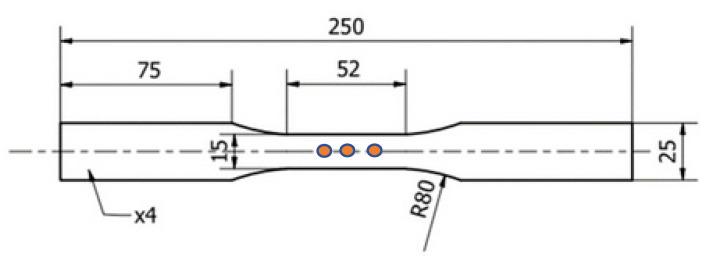
Geometry of the specimen used for uniaxial tensile and standard fatigue tests with marked areas of EC measurements (units in mm).

**Figure 3 materials-14-03824-f003:**
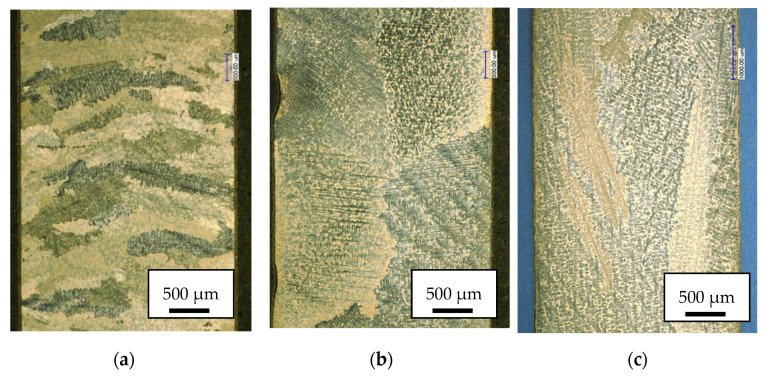
Microscopic structure of the MAR 247 nickel superalloy: (**a**) fine grain, (**b**) coarse grain and (**c**) column grain observed by using light optical microscope.

**Figure 4 materials-14-03824-f004:**
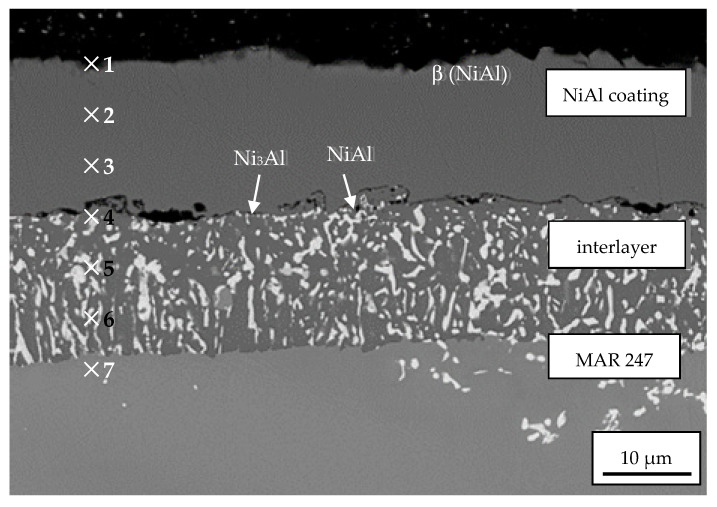
Microstructure of 40 µm NiAl coating produced by the CVD method on the MAR 247 nickel superalloy at 1040 °C with points of EDS analysis marked on crack propagation area.

**Figure 5 materials-14-03824-f005:**
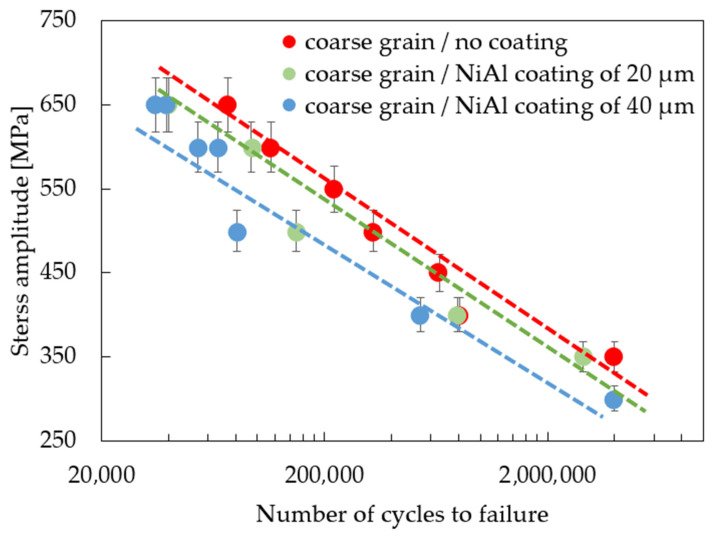
S-N curves for the coated, coarse-grained MAR 247 nickel-based superalloy.

**Figure 6 materials-14-03824-f006:**
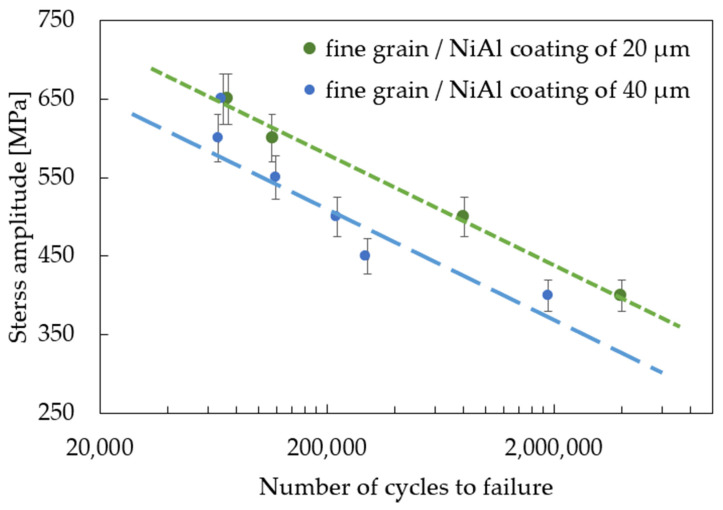
S-N curves for the coated, fine-grained MAR 247 nickel-based superalloy.

**Figure 7 materials-14-03824-f007:**
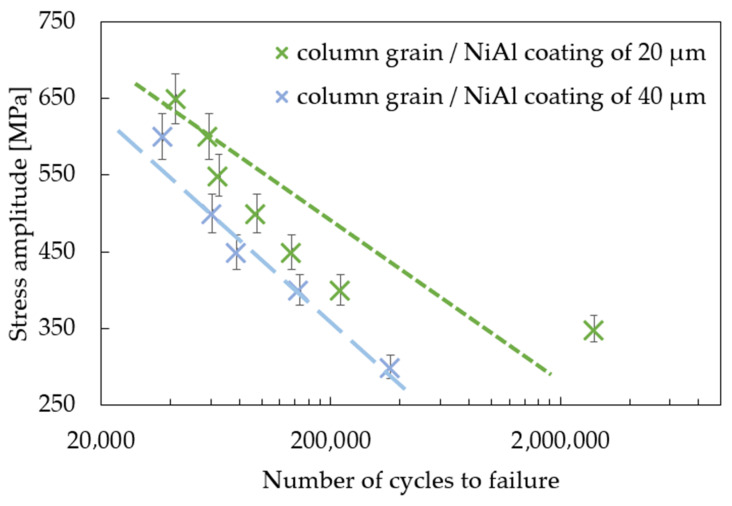
S-N curves for the coated, column-grained MAR 247 nickel-based superalloy.

**Figure 8 materials-14-03824-f008:**
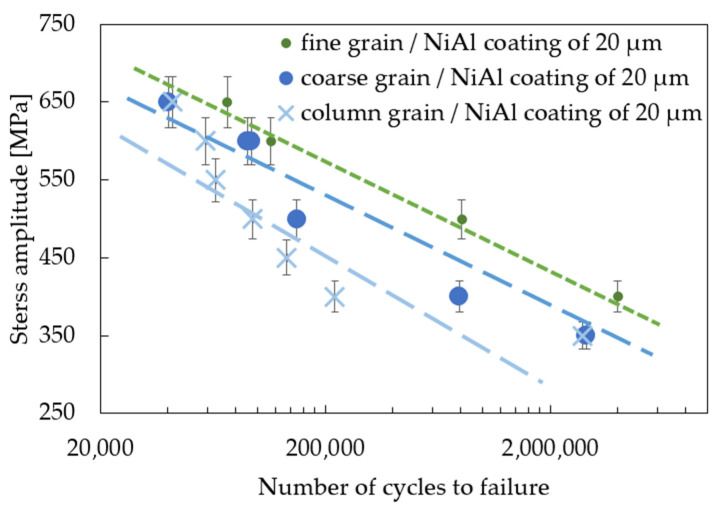
S-N curves for the MAR 247 nickel-based superalloy coated with 20 µm-thick NiAl.

**Figure 9 materials-14-03824-f009:**
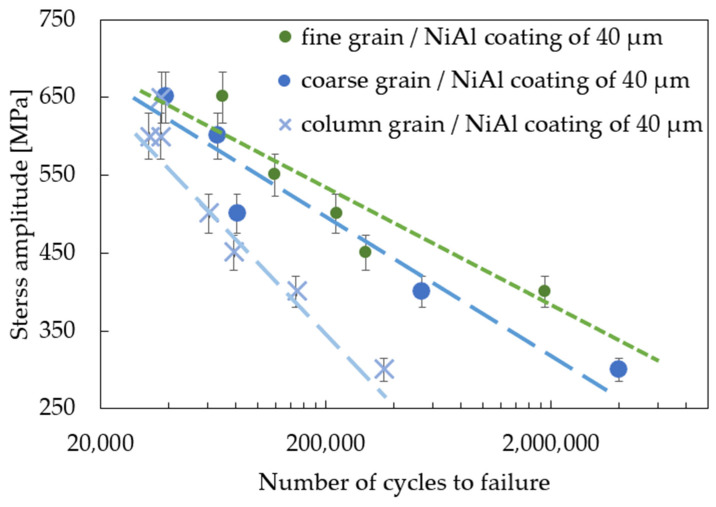
S-N curves for the MAR 247 nickel-based superalloy coated with 40 µm-thick NiAl.

**Figure 10 materials-14-03824-f010:**
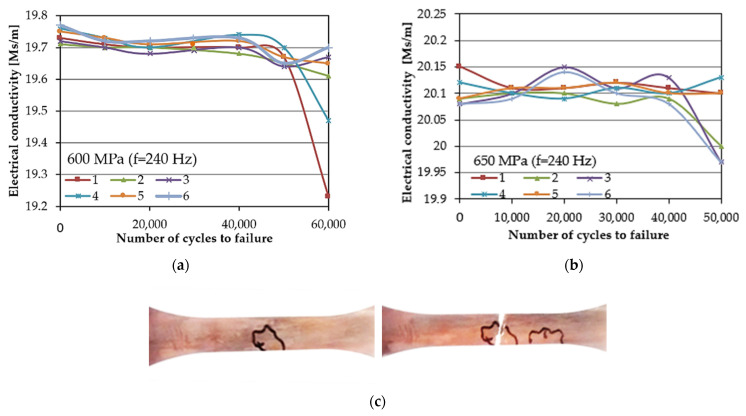
The exemplary results of electrical conductivity evolution measured in the selected loading cycles for two different levels of loading (**a**,**b**). Localized area of the potential crack and fracture of the specimen found after fatigue test (**c**).

**Figure 11 materials-14-03824-f011:**
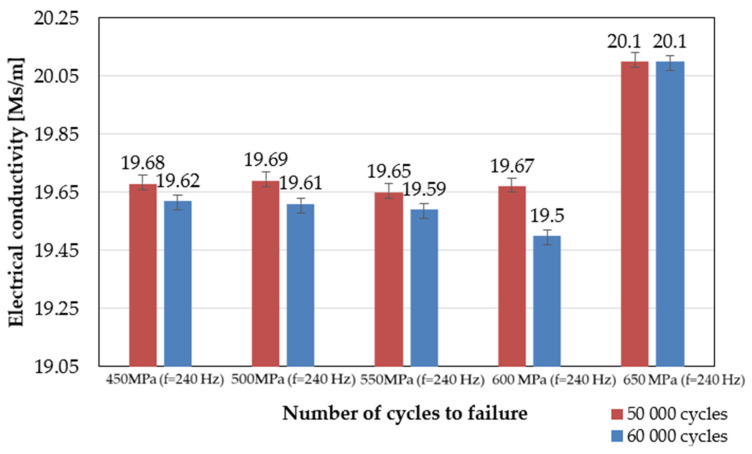
Comparison of electrical conductivity evolution measurements in the selected loading cycles.

**Figure 12 materials-14-03824-f012:**
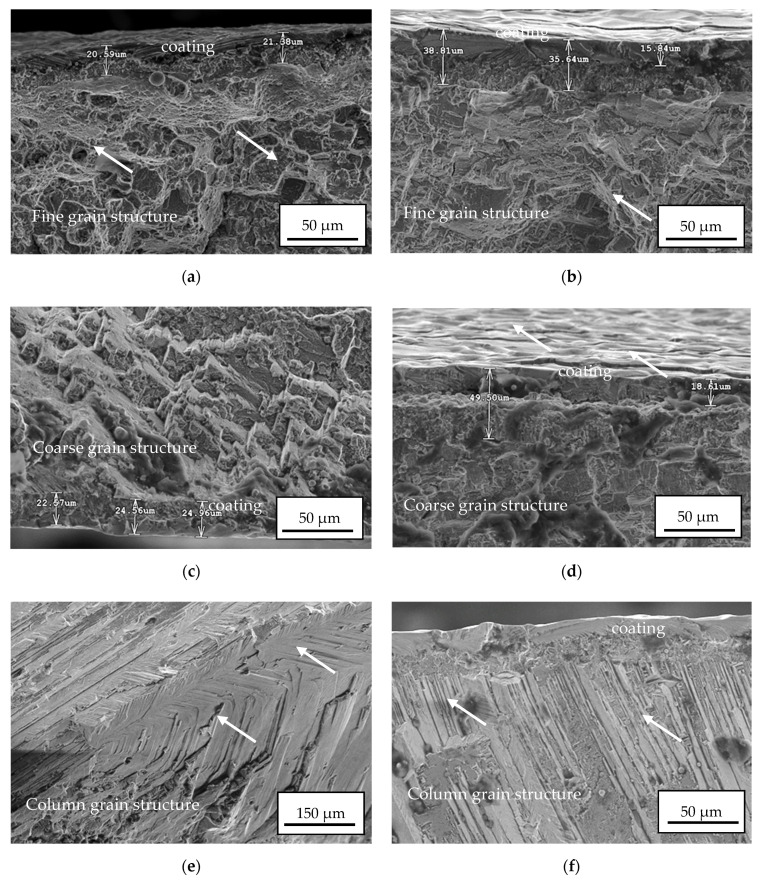
Fracture surfaces of the coated MAR 247 nickel-based superalloy: (**a**) fine grain/NiAl coating of 20 µm; (**b**) fine grain/NiAl coating of 40 µm; (**c**) coarse grain/NiAl coating of 20 µm; (**d**) coarse grain/NiAl coating of 40 µm; (**e**) column grain/NiAl coating of 20 µm; (**f**) column grain/NiAl coating of 40 µm observed by using scanning electron microscope.

**Table 1 materials-14-03824-t001:** Chemical composition of MAR 247 superalloy (wt. %).

C	Cr	Mn	Si	W	Co	Al	Ni
0.09	8.80	0.10	0.25	9.70	9.50	5.70	bal.

**Table 2 materials-14-03824-t002:** Chemical composition (wt. %) of coating surface obtained after deposition at 1040 °C.

Point	Al	Si	Ti	Cr	Co	Ni
1	25.01	x	x	2.66	7.78	64.55
2	23.68	x	x	3.23	7.95	65.15
3	21.79	x	0.49	4.63	8.69	64.41
4	19.31	0.34	0.89	6.52	9.57	63.00
5	18.73	0.51	1.07	5.95	9.84	63.64
6	17.41	0.53	1.20	7.07	9.44	63.98
7	6.47	0.23	0.65	7.21	9.71	59.85

## Data Availability

Data available in a publicly accessible repository.

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
