# Peer review of "Nondestructive Methodology for Identification of Local Discontinuities in Aluminide Layer-Coated MAR 247 during Its Fatigue Performance"

_materials, 2021, doi:10.3390/ma14143824_

Round 1

Reviewer 1 Report

The paper is well written. I don't have any further questions. 

Author Response

The authors would like to thank the reviewer for his time.

Reviewer 2 Report

I recommend that the editor accept this treatise as it is.

Author Response

(The authors gave the same response as above.)

Reviewer 3 Report

Comments of materials-1254196

The main weaknesses of the manuscript:

  1. Room temperature is not meaningful, therefore it is better to specify the exact temperature in °C.
  2. The figure captions of Fig. 4: “(d)” should be deleted.
  3. 12: The picture comment is not clearly stated, please add the auxiliary instructions (such as text, lines, pattern, etc.). Make it easy for readers to read.

Author Response

The authors would like to thank the reviewer for his time and comments. We have addressed all of the comments and revised the manuscript accordingly. All of the changes have been highlighted in yellow in the revised manuscript. Figure 12 was improved as well.

Reviewer 4 Report

This manuscript tested the fatigue performance of the aluminide layer coated nickel superalloy with three different initial microstructures using nondestructive, eddy current method. It turns out that nickel superalloy with a thinner coat have a better fatigue performance. Besides, fine grain superalloy has the best fatigue compared to coarse and column- structured grains. Also, they tested the electrical conductivity change to monitor the potential crack initiation and its propagation. In this manuscript, the experiments and results were well presented. In my opinion, this paper can be published in current form after carefully check.

  1. In line 72, I think 300℃ should be 300K.
  2. In line 263, I can’t find fig.10d

Author Response

Response:

Thank you for your review.

  1. In line there should be 300℃(reference: 1. Curry, N., Markocsan, N., Li, XH. et al. Next Generation Thermal Barrier Coatings for the Gas Turbine Industry. J Therm Spray Tech 2011 20, 108–115 https://doi.org/10.1007/s11666-010-9593-x)
  2. There should be Figure 10c. Thank you. The mistake was corrected.

Reviewer 5 Report

The authors of the present investigation have reported results of the fatigue performance of  as received MAR 247 alloy, i.e., Nickel super alloy, and a coated MAR 247 alloy by using the non-destructive technique of Eddy current for detecting discontinuities, i.e., defects, cracks.    An aluminide layer (AlNi) coating was applied on the NAR 247 samples by using the chemical vapor deposition (CVD) method.   Two layers of 20 um and 40 um were deposited on the MAR 247 samples at 1040 oC.   The as received MAR 247 samples were initially prepared to have fine, course, and columnar micro-structured grains.    The microstructures of the as received and the coated MAR 247 sample were characterized by optical microscopy and energy dispersive spectroscopy (EDS).   Fatigue tests were conducted on all the prepared MAR 247 samples to obtain the Stress-Number of cycle (SN)plots.   Then, two stress amplitudes of 450 MPa and 600 MPa were saparetly applied in a similar frequency of 240 Hz on all samples to determine the discontinuities in the microstructures of the samples on a predetermined number of cycle, each 10000 cycle, by the  EC technique.

In general the results show of the present study that the best fatigue performance was achieved by the as the received MAR 247 samples, then the coated MAR 247 with fine, course, and columnar grained microstructures.  Also, it has been determined that the lower the thickness of the aluminide layer the better the fatigue performance.   Furthermore, examinations of the scanning electron microscopy(SEM) to the fracture surface of the tested MAR 247 samples indicated that the as received MAR 247 and the coated MAR 247 with fine grained microstructure were fractured through the ductile fracture mode.   In contrast, the SEM examinations clearly indicated that the coated MRA 247 with course and columnar grained microstructures were fractured with the brittle fracture mode.       

The following items need to be revised:

1-In page 4 of the text, at line 35, the statement "an the optimization" should be revised to "the optimization".

Author Response

(The authors gave the same response as above.)
